# PROTACs and Building Blocks: The 2D Chemical Space in Very Early Drug Discovery

**DOI:** 10.3390/molecules26030672

**Published:** 2021-01-28

**Authors:** Giuseppe Ermondi, Diego Garcia-Jimenez, Giulia Caron

**Affiliations:** Molecular Biotechnology and Health Sciences Department, University of Torino, Via Quarello, 15, 10135 Torino, Italy; giuseppe.ermondi@unito.it (G.E.); diego.garciajimenez@edu.unito.it (D.G.-J.)

**Keywords:** chemical space, degrader, 2D physicochemical descriptors, permeability, PROTAC

## Abstract

Targeted protein degradation by PROTACs has emerged as a new modality for the knockdown of a range of proteins, and, more recently, it has become increasingly clear that the PROTAC chemical space requires characterization through a pool of ad hoc physicochemical descriptors. In this study, a new database named PROTAC-DB that provides extensive information about PROTACs and building blocks was used to obtain the 2D chemical structures of about 1600 PROTACs, 60 E3 ligands, 800 linkers, and 202 warheads. For every structure, we calculated a pool of seven 2D descriptors carefully identified as informative for large and flexible structures. For comparison purposes, the same procedure was applied to a dataset of about 50 bRo5 approved drugs reported in the literature. Correlation matrices, PCAs, box plots, and other graphical tools were used to define and understand the chemical space covered by PROTACs and building blocks in relation to other compounds. Results show that linkers have different properties than E3 ligands and warheads. Polar descriptors additivity is not respected when passing from building blocks to degraders. Moreover, a very preliminary analysis based on three PROTACs with high, intermediate, and low permeability showed how the most permeable compounds seem to occupy a region closer to bRo5 drugs and, thus, exhibit different properties than impermeable compounds. Finally, a second database, PROTACpedia, was used to discuss the relevance of physicochemical descriptors on degradation activity.

## 1. Introduction

Historically, biological targets are most commonly receptors, enzymes, ion channels, and transporters. However, in recent years, new drug targets are emerging and many diseases that had an unknown etiology have now been revealed to have new and difficult-to-treat drug targets. These targets show several impediments for drug discovery, such as inaccessible localizations (intracellular or nuclear) and the lack of deep pockets to bind drugs needed for their modulation [1,2]. Notably, they have been often related to protein dependent diseases, whose main treatment scopes act at the transcriptional (CRISPR/Cas9) or the post-translational level [3,4]. Under physiological conditions, post-translational level protein regulation mostly makes use of the ubiquitin-proteasome system (UPS). In fact, it was in 2001 that Crews and Deshaies designed the first hetero-bifunctional molecules that used UPS-mediated protein degradation to act as drugs [5,6]. Their structure consists of three building blocks: a ligand that binds a protein of interest, a ligand that recruits an E3 ubiquitin ligase, and a linker that attaches both regions. These large molecules (about 700–1100 Da) are widely known as degraders, even though the commercial name is PROTAC^®^ (PROteolysis Targeting Chimera) [7,8]. Despite issues in the identification of the optimal method and/or methods combination to assess protein degradation efficacy (biochemical, biophysical, and cellular techniques are available [9]), which is compared to traditional inhibitors. Degraders are more potent and selective, sustain longer effects, and work at lower doses [10]. Moreover, PROTACs have a wide range of potential applications. For instance, proteins that have been successfully degraded by PROTACs include nuclear receptors, protein kinases, epigenetic regulators, neurodegenerative disease-related proteins, virus-related proteins, and transcription factors [11].

However, regardless of their pharmacological action, degraders often show bioavailability limitations related to drug metabolism and pharmacokinetic (DMPK) challenges. Degraders belong to a category of compounds known as beyond Rule-of-5 (bRo5) that does not comply with the Lipinski’s rule of 5 (Ro5) [12]. However, the bRo5 chemical space includes several commercial drugs with good bioavailability [13]. Therefore, an analysis of the degraders’ chemical space is extremely relevant to understand more about their potential as drugs. This was also very recently highlighted by Troup et al. in their excellent review [14], which supports the relevance of physicochemical properties in the PROTAC drug optimization process. Moreover, two pivotal studies specifically focused on this topic have been published in the last two years. Edmondson et al. in 2019 [15] evaluated several physicochemical descriptors of about 40 PROTACS including molecular weight (MW), lipophilicity (cLogP, cLogD), H-bond donor (HBDs), and H-bond acceptor (HBAs) counts, polar surface area (PSAs), the number of rotatable bonds (nRotB), and the AbbVie score named AB-MPS [16]. Maple et al. in 2019 [17] manually set-up a comprehensive database of 422 degraders retrieved from the literature and provided an overall summary of the physicochemical parameters. Moreover, the authors also compared the physicochemical profiles of efficacious PROTACs with other published sets of bRo5 compounds.

Although very recently published, we reasoned that at least two main arguments justify an update of these two studies. First, the number of publications reporting new PROTAC structures is exponentially increasing and two PROTAC-based databases are available today (PROTACpedia, http://protacdb.weizmann.ac.il/ptcb/stats and PROTAC-DB, http://cadd.zju.edu.cn/protacdb/ [18]). Notably, these tools represent the main source of information about this new class of compounds and the availability of free resources has been shown in the past to significantly promote the development of the considered research field. The PDB database (https://www.rcsb.org/), ChemSpider (http://www.chemspider.com/), DrugBank (https://go.drugbank.com/), and ZINC (https://zinc.docking.org/) are examples of public databases commonly used in drug discovery studies. The Ro5 itself derives from the availability of a large database of compounds, including experimental data related to drug absorption. The second reason for a new analysis of PROTACs physicochemical descriptors is the evidence that a careful selection of ad hoc descriptors is required to monitor bRo5 properties. For instance, a few studies showed that traditional in silico lipophilicity prediction may fail when degraders are considered [19,20]. Moreover, flexibility should be considered with great attention as well, mainly when molecules including cyclic moieties are investigated [21].

Therefore, we designed a study aimed at three main objectives: (a) highlighting the available resources to retrieve PROTAC-related structures, (b) providing a reasonable pool of computed 2D descriptors and discuss their pros and cons in the characterization of the PROTACs chemical space, and (c) exploring PROTACs chemical space in relation with their building blocks. To reach our aims PROTACs, warheads, E3 ligands, and linkers were downloaded and a pool of seven 2D descriptors was calculated for all the compounds (molecular weight = MW, the number of carbon atoms = nC, the number of the aromatic rings = NAR, the Kier’s flexibility index = PHI, the number of acceptor atoms for H-bonds = nHAcc, the number of donor atoms for H-bonds = nHDon, and the topological polar surface area = TPSA). Chemometric tools and graphical data analysis were then applied to check the variation in descriptors inside the different classes and the impact of building blocks on PROTACs properties. Moreover, a comparison with a set of 52 bRo5 compounds already reported in the literature [13,22] was also performed and a preliminary analysis about the potential identification of the chemical space regions were occupied by permeable derivatives as well. Finally, PROTACpedia was also used to discuss the relevance of physicochemical descriptors on degradation activity.

## 2. Results and Discussion

### 2.1. Descriptors Selection

Generally speaking, molecular descriptors used to characterize drug-like compounds are those implemented in common metrics, such as Lipinski, Veber, etc. The main advantage of these descriptors is their 2D nature that enables a fast calculation and, thus, their application in very early drug discovery. A second advantage is their simplicity that allows their straightforward interpretation and use by medicinal chemists. Nevertheless, some of them exhibit drawbacks often masked in the Ro5 space (see below).

Figure 1 classifies physicochemical descriptors in three main classes (2D, 3D, and experimental) that are expected to be used at different phases of early drug discovery. In the very beginning when a huge number of virtual compounds are under study, the use of 2D descriptors is mandatory. However, we can distinguish two types of 2D descriptors: pure structural descriptors (e.g., MW) and descriptors that predict experimental data (e.g., log P). 3D descriptors need the building of at least one conformer of the compounds even though it is better to have a pool of conformers of interest. In practice, they can be used in a later stage of development when the number of virtual compounds is reduced and, thus, more time and expertise requiring tools can be applied. Finally, once some compounds are synthesized and experimentally tested, it is possible to measure experimental descriptors (mostly obtained through chromatography due to suboptimal compound purity [23]).

For this study on PROTACs, we focused on pure structural 2D descriptors to provide a univocal starting point in the definition of degraders’ chemical space to be used in very early drug discovery. We decided to privilege a constructionist rather than a reductionist approach often associated with the inclusion of too many not completely clear descriptors. It seems clear that the number of descriptors is expected to be increased in the near future once an acceptable number of experimental data will be collected. For instance, new determinants able to consider the potential of degraders to form intramolecular hydrogen bonds are expected to be included in an updated version of the selected descriptors.

Despite the huge number of free and commercial log P calculators, none of them can properly predict bRo5 and, thus, PROTAC log P. Since enough validated experimental log Ps are not available, log P is not included in the set of descriptors used to develop the method. A few examples of failure were reported in the literature [19,20]. Calculation of log D is even worse because of the need to predict pK_a_ that suffers from the same problem. It could be stated that a relative comparison of calculated log P values could be beneficial for a series of similar degraders. This observation is true but related to specific situations and, thus, supports our suggestion of not including lipophilicity descriptors in this study. A second major point regards TPSA, which is a polarity descriptor. TPSA has no relation with experimental data and is a pure structural descriptor. Although TPSA does not allow us to distinguish whether a fragment is accessible to solvent or not, we decided to include TPSA in this study since it provides a picture of the maximum polarity that can be expressed by a molecule [24,25] and, as such, has a value. A third alert about the use of common descriptors in the bRo5 chemical space concerns hydrogen bond counts (i.e., hydrogen bond donor and hydrogen bond acceptor properties). In fact, we need to point out that the number of HBD/HBA groups may be different from the number of HBD/HBA atoms. Some software does not declare which descriptor is computed and this fact could represent a source of errors when comparing literature values. In this paper, the number of atoms is used to compute HB properties. Like TPSA, HBD/HBA counts are related to the maximum number of HBD/HBA that can be expressed by a molecule. Finally, we need to focus on flexibility descriptors. In a very recent study [21], we clearly outlined that, in the bRo5 space, the Kier’s flexibility index (PHI or F) should be preferred over the number of rotatable bonds (RBN or NRot) since PHI takes into account the flexibility of macrocycles neglected by RBN by definition.

According to the previously mentioned discussion, here, we decided to use the following descriptors (Figure 2 and Appendix A for definition): MW as a fast-calculated descriptor of molecular size, a set of count descriptors related to both polar (nHAcc (also HBA), nHDon (also HBD)), and nonpolar (the number of carbon atoms, nC and the number of aromatic rings, NAR) molecular moieties, a flexibility descriptor (PHI) and TPSA as a polarity index. In practice, we set up a pool of seven descriptors, three of a nonpolar nature (in yellow in Figure 2), three of a polar nature (light blue in Figure 2), and one flexibility descriptor (in violet, Figure 2).

We are aware of the arbitrariness in the choice of a small set among the dozens of existing interpretable descriptors, but we think that, at this stage of knowledge on degraders, a simple view based on carefully selected physicochemical properties can be more useful for researchers working in the field than a complex net of difficult-to-interpret descriptors. Moreover, in the S.I., we reported a statistical analysis to further support our choice.

Notably, our pool of descriptors takes also into account the components of molecular lipophilicity. In fact, it has been shown [26,27] that both nonpolar and polar descriptors are used to define the bRo5 chemical space. nC and NAR are related to the nonpolar portion of the molecule, (hydrophobicity-related, Equation (1)) whereas TPSA, nHAcc, and nHDon are related to the polar components (polarity-related, Equation (1)). We are clearly aware about the possibility that, in principle, some carbon atoms can bring a partial positive charge due to the link with more electronegative atoms, e.g., carbonylic groups. However, the same argument can be extended to the definition of TPSA where only nitrogen and oxygen atoms are considered in the count. Therefore, nC can be considered a choice of nonpolar descriptors coherent with the TPSA definition.
*lipophilicity* = *hydrophobicity* − *polarity*(1)

Notably, the chosen descriptors can be calculated for the PROTACs and the building blocks with the same accuracy. Thus, they can be compared without the bias introduced by predicted values based on experimental data that can be influenced by the applicability domain of the model. Overall, the selected pool of seven descriptors is uniquely defined and does not depend on experimental values that are not available for this new class of compounds. They are, therefore, expected to be used in the very early drug discovery (Figure 2), when researchers have to deal with many virtual and not very well-known compounds like degraders.

### 2.2. Databases and Datasets

The science behind degraders is still very recent and complex compared to traditional molecules, either designed in the Ro5 chemical context or obtained from a biological origin. Thus, widely used chemoinformatic resources, such as PubChem (https://pubchem.ncbi.nlm.nih.gov/), still offer limited, detailed knowledge. Recently, databases completely specific for degraders are being developed. For instance, PROTACpedia, a collaborative and freely accessible resource of manually curated data (http://protacdb.weizmann.ac.il/ptcb/main), redirects to the original literature supporting a degrader, warhead, or E3 ligase search. Furthermore, a more comprehensive database named PROTAC-DB (http://cadd.zju.edu.cn/protacdb/) [18] provides first-hand information about chemical structures, biological activities, some physicochemical properties, and the corresponding references. In PROTAC-DB, the biological activities contain the degradation capacities, binding affinities, and cellular activities. However, many data are missing, and no activity has a value for all (or most of) degraders. PROTACpedia reports a tag (active/inactive) based on the report given in the appropriate paper. Notably, both databases, which are a valuable resource in the study and development of degraders, do not report any experimental data of lipophilicity and permeability. However, as demonstrated by the Ro5, the availability of these data could be pivotal to model degraders’ bioavailability and, thus, their future as drugs.

The two databases include a different number of PROTACs entries (1662 in PROTAC-DB and 779 in PROTACpedia). Appendix A shows a very high overlap between the two PROTAC sets and, thus, due to the larger number of entries, PROTAC-DB was considered for most of the further analyses. All entries including PROTACs, warheads, E3 ligands, and linkers were downloaded from PROTAC-DB as either SDF or CSV files [18]. Figure 3A summarizes the number of compounds available for every class. For the sake of clarity in the paper, we used the following color codes: E3 ligands = red, linkers = green, warhead = blue, and PROTACs = grey. Most of the building blocks are Ro5 compliant (but 46 warheads, 3 E3 ligands, and 3 linkers) whereas, as expected, PROTACs are bRo5 compounds. Overall, the investigated PROTACs were not necessarily intended to be dosed orally. Rather they were either cellular tools or in vivo tools intended to validate PROTAC biology. A detailed analysis of PROTAC-DB entries is reported in Appendix A (E3 ligands) and S3 (warheads). Overall, the E3 ligase enzymes harnessed by the degraders in the dataset belong to eight family classes whereas 59 classes of proteins of interest are considered. The 806 linkers include both aliphatic and aromatic derivatives. The high number of linkers could be due to the evidence that a significant proportion of degraders were developed through mostly empirical optimization of linker composition, which usually requires the synthesis of large libraries of compounds containing linkers of various natures [14]. A second consideration on linkers is related to the different functionalization at either end [14]. A detailed analysis of the sites conjugating the linker with warheads and E3 ligands is beyond the scope of our study. However, we verified how the database handles the junction sites. For instance, MZ1 and its building blocks are displayed in Figure 3B, as provided by the PROTAC-DB. The database follows the same rational methodology to cleave every degrader. It displays the simplest chemical moieties required to build the PROTAC, setting as R_1_ and R_2_ the connections from the warhead to the linker and the linker to the E3 ligase, respectively. Even though this structural fragmentation undoubtedly helps to design and retrieve the fundamental building blocks involved in each PROTAC, it does not efficiently consider the synthetic methodology used to connect each fragment. For example, as shown for MZ1 in Figure 3B, the E3 ligase (DB ID = 12) contains an amide group that connects to the linker (DB ID = 114). However, as described by Ciulli et al. [28], the synthetic scheme can involve the use of an E3 ligase containing a terminal amino group (highlighted with the blue circle in Figure 3C) that reacts with the carboxylic acid in the linker (highlighted with the red circle in Figure 3C). Therefore, the synthetic requirements and commercial availability of reagents suggest that the fragment proposal presented by the database is not representative enough.

### 2.3. Analysis of Chemical Space Distribution of PROTACs and Their Building Blocks

The seven descriptors were calculated for all the downloaded compounds. Correlation matrices are reported in Figure 4 and show similar but not identical behavior among the four classes (i.e., PROTACs, E3 ligands, warheads, and linkers) of compounds.

For any class, TPSA is significantly correlated with all descriptors but NAR. As expected, a high correlation with nHAcc and nHDon is verified due to the calculation of TPSA that takes into account nitrogen and oxygen atoms, i.e., the atoms with HB properties. Remarkably, there is a positive correlation with MW and in a smaller extent to nC. This is not clear and means that the increase in molecular size is often achieved with moieties that include polar atoms. One reason that could explain this observation is likely related to the exigence in the PROTAC synthesis of reactive groups that often include nitrogen and oxygen atoms. Another point could be the introduction of polar groups during the optimization process of the warhead to obtain stronger binding with the protein of interest. nHDon is not significantly related to an increase of MW, nC, and NAR. In practice, an increase in dimensions does not induce an increase in nHDon. nHAcc has a different behavior since it increases with MW. This is reasonable since any nHAcc group may also be the HB donor, but nHDon are also nHAcc. Lastly, PHI is mainly correlated with MW and nC.

When linkers are considered (Figure 4C), PHI also shows a weak negative correlation with NAR, which suggests that flexibility decreases with the increase of the number of aromatic rings. NAR is mainly related to MW and nC as expected. The correlation found with nHAcc in warheads (Figure 4B) could be related to the presence of heterocycles. MW is highly correlated with nC, PHI, and nHAcc but not with NAR and nHDon, which are poorly correlated with any descriptor.

Overall, it seems that, when increasing the PROTAC dimensions, there is an increase in polarity and it is not clear if this is either a design or a synthetic need. From a drug discovery point of view, this is a major point since the increase in polarity is often related to a decrease in permeability.

The next step of the analysis consisted in looking for compounds with similar properties (i.e., clusters). Principal Component Analysis (PCA) was then performed, where PCA is a dimensionality-reduction method that reduces the dimensionality of the descriptors, by transforming a large set of variables (the descriptors) into a smaller one that still contains most of the information in the large set. The score and loading plots are in Figure 5. Briefly, the score plot permits us to identify clusters of molecules, whereas the loading plot shows which descriptors have the largest effect on each component.

The variance explained by PC1 and PC2 in the different classes is rather similar (Appendix A) and explains between about 70% of the total variance (linkers and PROTAcs) and 85% (E3 ligands and warheads). With the inclusion of the PCA3, the pool of descriptors explains about 89% of the PROTAC variance (and about 93% of the building blocks). The loading plots of warheads (Figure 5B) are quite similar to those of PROTACs (Figure 5D) and not too different in terms of a descriptor location (score plot, Figure 5A). In these three plots, polar descriptors are located in the opposite quadrant than the hydrophobic descriptors with PHI in the middle. The linkers behave differently from other compounds as revealed by both the score and the loading plots. The score plot reveals that NAR and PHI are the most important descriptors of PC2 and nHDon is less important than other descriptors. Furthermore, NAR and PHI are located in PC2 in opposite directions, suggesting that the increase of NAR reduces the flexibility of the molecule. The loadings plot shows that they can be divided into three main classes. The analysis of the structures reveals that the lower class includes linkers with two aromatic rings, the middle-class includes linkers with one aromatic ring, and the most populated class includes links with no aromatic rings. Particularly, the different junction sites do not make the linkers very different from one another. Therefore, these findings should be taken into account in optimizing the linker moiety for the modulation of physicochemical parameters and, ultimately, controlling the drug metabolism and pharmacokinetic (DMPK) profiles of PROTAC degraders.

Overall, PCA shows that linkers behave differently from other compounds and can be grouped in three classes, according to the number of aromatic rings.

PROTACs are obtained by merging three building blocks with different properties. The resulting compounds are clearly new entities with new properties, but it could be interesting to understand which properties are maintained and/or how the block properties influence the new entity. This information can be useful in the design of PROTACs. A box plot is a non-parametric method for graphically depicting groups of numerical data through their quartiles. These plots were built for the four classes (E3 ligands, linkers, warheads, and PROTACs) to compare properties distribution among them. Figure 6 focuses on nonpolar descriptors. In the case of MW, the sum of the medians of the building blocks gives a slightly larger value than the PROTAC median (Appendix A). This result is in line with the fact that PROTACs molecules are obtained by merging blocks that lose some functional groups to form the final product. The same is true for nC, whereas, for NAR, the PROTAC value is exactly the sum of others. A larger number of aromatic rings are present in the warheads likely because they are required for an efficient binding with the protein of interest. The presence of aromatic rings in warheads has minor influence on the flexibility of the final product. This is in line with PCA where NAR and PHI are orthogonal (Figure 6C). The larger number of aromatic rings of warheads supports the fact that the nonpolar properties of PROTACs mostly depend on them.

Figure 7 shows the trend for polar descriptors. For TPSA, additivity is not respected likely because the formation of degraders from building blocks causes the elimination of some polar groups. nHDon is less addictive than nHAcc. E3 ligands and warheads seem to contribute similarly to polar properties of PROTACs (Figure 7). Notably, nHDon of PROTACs are in line with Lipinski, as already observed by Maple et al. [17].

Finally, we focused on flexibility (PHI, Figure 8). It is not additive since PROTACs are largely more flexible than the sum of the medians of building blocks. Moreover, PROTACs’ flexibility could be mainly governed by the linkers, which show a larger flexibility range than E3 ligands and warheads even though there are several poorly flexible linkers (a significant number of outliers with a PHI value lower than PHI mean). As remarked above, the warhead has little influence on flexibility despite the presence of a larger number of aromatic moieties. PHI shows how the flexibility of the final PROTAC requires careful analysis. The final molecule is more flexible than the corresponding blocks, but it is not the simple sum of them, as we can expect by descriptors that increase with the size of the molecule.

Overall PROTACs result as flexible molecules containing a hydrogen bond acceptor and donor groups that, at least in principle, can form intramolecular hydrogen bonds (IMHBs). Furthermore, these molecules are also rich in nonpolar moieties that can be involved in a sort of hydrophobic collapse, i.e., other intramolecular interactions. The size of the molecules suggests a more careful analysis of the properties related to the flexibility and the search for descriptors of the molecular propensity to sustain intramolecular interactions. A better understanding of the influence of the intramolecular interactions on the final PROTACs properties calls, however, for more complex descriptors not available in very early drug discovery.

### 2.4. PROTACs vs. the bRo5 Dataset of Approved Drugs

Kihlberg and coworkers recently published about 50 oral drugs that have been approved in the bRo5 space [13,22]. The seven descriptors described above were also calculated for these compounds (Appendix A). Mean values are in Figure 9A and support and extend what is already observed in previous studies about the larger chemical space occupied by PROTACs (which, in our study, includes unoptimized structures and not approved drugs). Outstandingly, nHDon is the only descriptor relatively similar between bRo5 and PROTACs. The six remaining descriptors show significantly larger values for PROTACs.

To visually compare datasets’ chemical space, we also built a 3D plot with three descriptors: nC (nonpolar), TPSA (polar), and PHI (flexibility) (Figure 9B). Linkers are the more flexible moieties and essentially form a cluster separate from other blocks. E3 ligands and warheads, on the other hand, are not separated and are shifted toward the bRo5 space. Finally, bRo5 drugs (in orange) behave like a sort of bridge between building blocks and degraders that form a new cluster apart.

### 2.5. Coming Soon: The Chemical Space of Oral Bioavailable PROTACs

As mentioned in the Introduction, PROTACs are expected to have poor bioavailability. Therefore, a very important challenge is to map the locations of bioavailable degraders (bioavailability is the fraction of an orally administered drug that reaches the systemic circulation) in the property space that we have previously characterized. To do as mentioned, bioavailability and/or in vitro property data like permeability, lipophilicity, and polarity are needed.

The bioavailability of a significant number of PROTACs has been discussed in recent papers by Astra Zeneca researchers [29] but, unfortunately, in vivo data are not available. Permeability is a major determinant of bioavailability but few permeability values of PROTACs have been reported in the literature. Moreover, the method to be used (cellular vs. artificial membranes) is still under debate. A recent paper [30] states that an efficient and reliable method to quantify the PROTAC cell permeability is missing, but Lokey and coworkers discussed PAMPA utility [31]. Overall, the feeling is that permeability determination for compounds with MW > 1000 could be an issue. This evidence supports a huge interest in physicochemical descriptors tailored to bRo5 compounds since lipophilicity and polarity have a huge potential to model bioavailability. Preliminary data by our group showed that two chromatographic descriptors quantifying lipophilicity in nonpolar environments (log k’80 PLRP-S) and polarity (Dlogk_w_^IAM^) could be of relevance in this respect [20]. We are working along these lines in our lab to extend the number of experimental descriptors to obtain useful data.

For a very preliminary investigation, here, we focused on three PROTACs with different Caco2 permeability values (high, medium, and low Caco-2 log P_app_) [20] and included in the PROTAC-DB: BI-3663 (log P_app_ = −4.8, ID = 1655), ACBI1 (log P_app_ = −5.7, ID = 798), and MZ1 (log P_app_ = −7.5, ID = 335), respectively (chemical structures in Appendix A). The selection of just these three compounds is due to the limited number of reliable and comparable permeability data reported in the literature. Figure 10A shows that MZ1 shows MW > 1000, a flexibility PHI, and a TPSA value significantly larger than the two permeable PROTACs. Excessive flexibility is often associated with poor permeability [32] and, therefore, it would be crucial to identify a filter value for flexibility. TPSA indicates that MZ1 may express a higher polarity than permeable degraders. PCA analysis on PROTAC and bRo5 drugs was finally performed. The loading plot (Figure 10B) is coherent with those reported in Figure 5 apart from linkers. The score plot (Figure 10C) shows an extended region in which both PROTACs and bRo5 drugs are present. Notably, the two permeable PROTACs are closer to the bRo5 zone than MZ1 (not permeable). Although more data should be obtained to confirm this very preliminary trend, this result supports the use of the seven descriptors for individuating oral degraders.

### 2.6. Degradation Activity and Physicochemical Descriptors

Published degraders range from inactive to very potent compounds. Therefore, in an ideal context, descriptors should be able to distinguish active compounds from inactive compounds. To discuss this aspect, all PROTACpedia entries were downloaded and divided in the two sets obtained by using the flag active/inactive (see above). Subsequently, the seven descriptors were calculated and a 3D plot with three representative descriptors nC (nonpolar), TPSA (polar), and PHI (flexibility) was built (Figure 11). No separation between active and not active compounds can be found.

This is an expected result from several points of view. First, biological data were obtained using very different methods (biochemical, biophysical, and cellular) and, thus, their values cannot be safely compared. For instance, some assays are in vitro (and, thus, not impacted by permeability issues, like ternary complex formation between the protein target, the PROTAC, and the recruited E3 ligase) and others are in cellular systems (like measuring targeting protein levels and ubiquitination). Moreover, the seven descriptors have been selected to monitor ADME-related properties and not binding events, like the formation of the ternary complex, which is considered paramount for successful degradation. In other words, these seven 2D descriptors are expected to fail with subsets of degraders for which coherent activity data are reported and different descriptors should be applied to monitor pharmacodynamic events.

## 3. Methods

The sdf and csv files were downloaded from both PROTAC-DB (last download on 10 November 2020) and PROTACpedia (last download on 28 December 2020). The csv files were transformed into xlsx files using Microsoft Excel (v. 16.0).

The sdf files were submitted to OSIRIS DataWarrior Version 5.2.1 (http://www.openmolecules.org/datawarrior/) and to alvaDesc 2.0.0 (https://www.alvascience.com/alvadesc/) without any further modification. Descriptors were calculated as follows: NAR were obtained with DataWarrior. All the remaining descriptors (Appendix A) were calculated with alvaDesc. Raw data were collected directly from the two software and pasted into the Excel files previously prepared.

Correlation matrices (Pearson) and PCA analysis were performed with alvaDesc using the default settings. The 2D scatter and box plots were obtained with Excel. 3D scatter plots were prepared with DataWarrior.

*p*-values for all correlations and ANOVA analyses were performed using Matlab 2019a.

## 4. Conclusions

Recent literature highlighted the need for physicochemical descriptors tailored to bRo5 compounds and online databases that collect chemical structures for a specific class of bRo5 compounds like degraders (and related building blocks). In this study, we set-up a pool of seven simple 2D structure-based descriptors and use them to define the chemical space of PROTACs and building blocks downloaded from PROTAC-DB and PROTACpedia. A set of about 50 approved bRo5 drugs was also studied for comparative purposes.

The analysis that allowed the selection of the final pool of descriptors clarified why, currently at least, lipophilicity descriptors cannot be included in the characterization of virtual degraders and recalled the significance of TPSA to avoid misinterpretation of its significance. Moreover, the need for the Kier’s flexibility descriptor in the bRo5 space is also remarked. The pool includes three nonpolar (nC, nC, and NAR) descriptors, three polar (TPSA, nHAcc, and nHDon) descriptors, and one flexibility (PHI) descriptor.

The correlation matrix, PCA, and the box-and-whisker plot show that linkers have different properties from E3 ligands and warheads and that the number of aromatic rings is crucial to identify linkers with different properties. Moreover, polar descriptors additivity is not respected when passing from building blocks to degraders and the average number of HB donor atoms is 4 ± 2. Finally, a very preliminary analysis based on three PROTACs with high, intermediate, and low permeability showed how the permeable compounds seem to occupy a region close to the bRo5 drugs and, thus, exhibit different properties compared to impermeable compounds.

Overall, this study describes the state-of-the-art of the 2D in-silico physicochemical descriptors that are required to characterize the chemical space of degraders in very early drug discovery. To individuate the regions occupied by bioavailable compounds and, thus, solid rules of thumb to apply in a filtering procedure, a decent number of experimental bioavailability/permeability/physicochemical data is mandatory and is expected to be obtained in the near future. Moreover, future studies should be focused on the design of a scoring index able to quantify the propensity of bRo5 compounds like PROTACs to form intramolecular interactions. In particular, an index specific for the prediction of intramolecular hydrogen bonds is of particular relevance and could ideally complete the pool of descriptors reported in this study. Work along these lines is in due course in our laboratories.

## Figures and Tables

**Figure 1 molecules-26-00672-f001:**
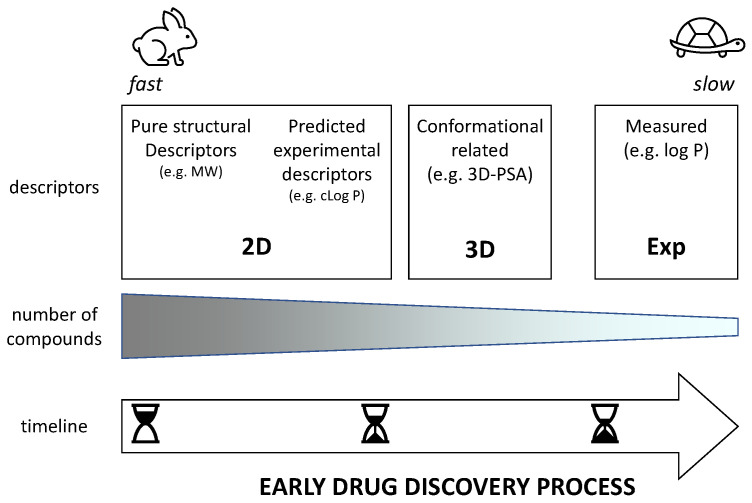
Physicochemical descriptor classification.

**Figure 2 molecules-26-00672-f002:**
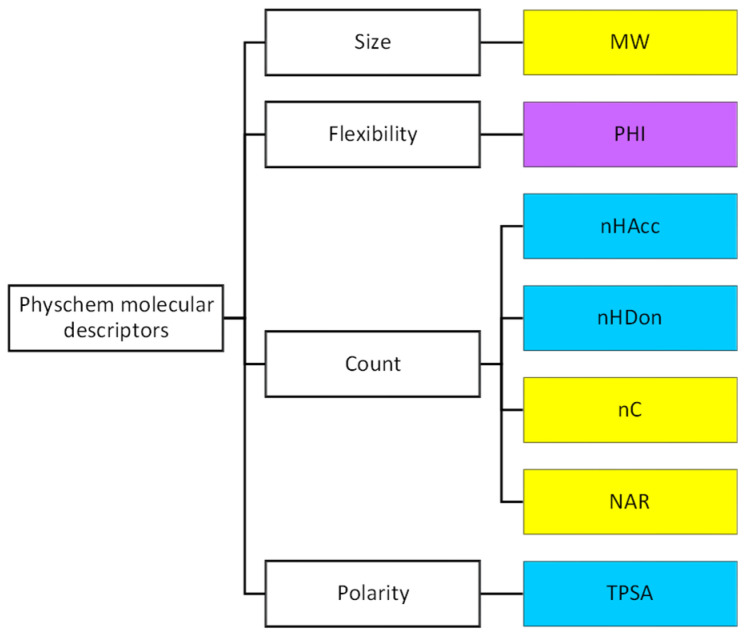
Molecular descriptors used in this study (yellow: descriptors related to hydrophobicity, light blue: descriptors related to polarity, violet: flexibility descriptors).

**Figure 3 molecules-26-00672-f003:**
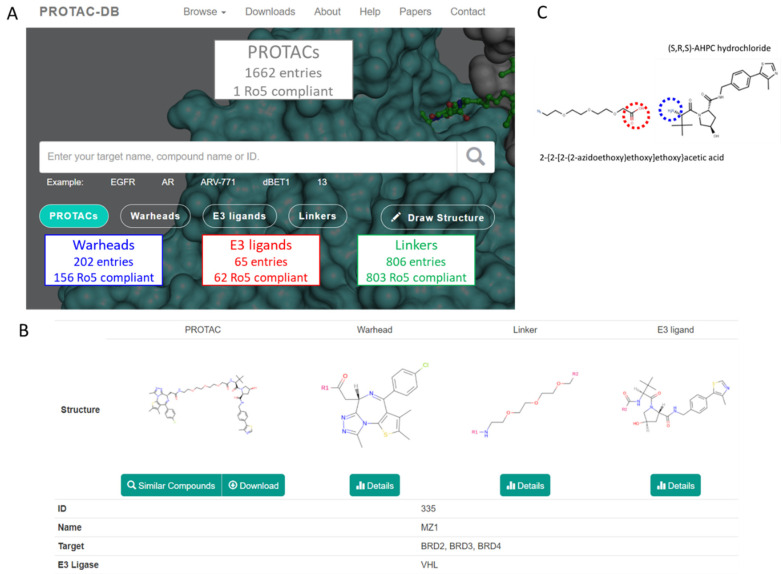
PROTAC-DB. (**A**) An overview of the included compounds, (**B**) the result of a Search procedure using MZ1 as an input, and (**C**) two building blocks effectively used to synthetically access the linker-E3 ligase substructure of MZ1 (see text for more details).

**Figure 4 molecules-26-00672-f004:**
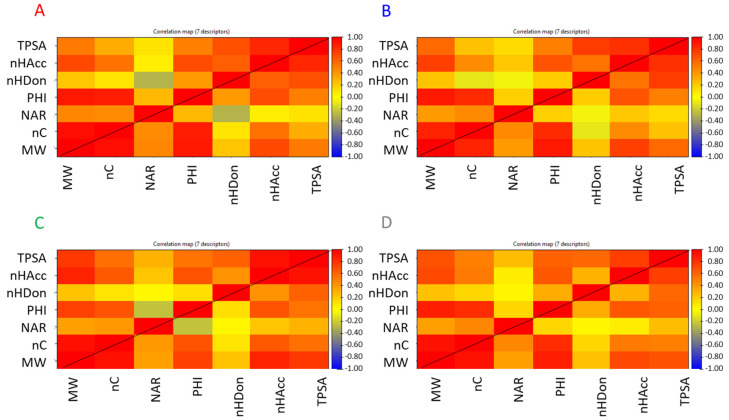
Correlation matrices. (**A**) E3 ligands, (**B**) warhead, (**C**) linkers, and (**D**) PROTACs. Pearson’s correlation coefficients and corresponding *p*-values are reported in Appendix A.

**Figure 5 molecules-26-00672-f005:**
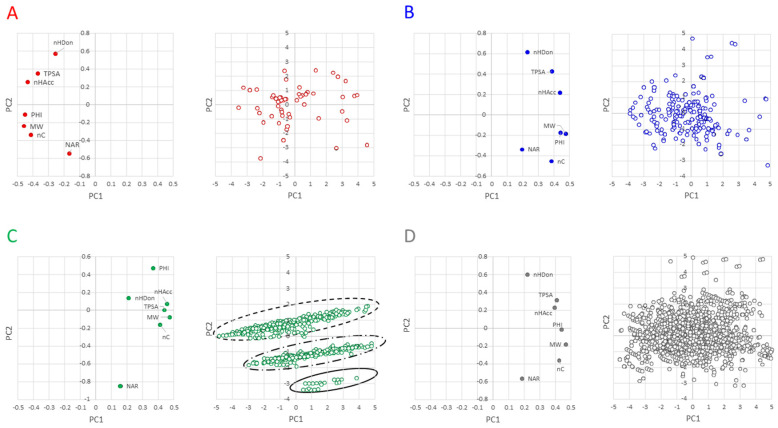
PCA. (**A**) E3 ligands (PC1 = 59.1%, PC2 = 25.0%), (**B**) warhead (PC1 = 56.9%, PC2 = 24.2%), (**C**) linkers (PC1 = 59.9%, PC2 = 17.4%) and (**D**) PROTACs (PC1 = 59.7%, PC2 = 17.4).

**Figure 6 molecules-26-00672-f006:**
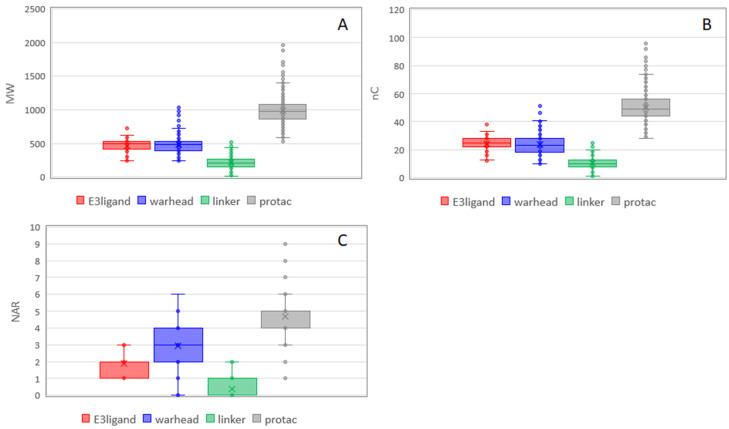
Box plots of the nonpolar descriptors for PROTACs and their building blocks. (**A**) MW, (**B**) nC, and (**C**) NAR. Statistical analysis was performed using ANOVA and *p*-values are reported in Appendix A.

**Figure 7 molecules-26-00672-f007:**
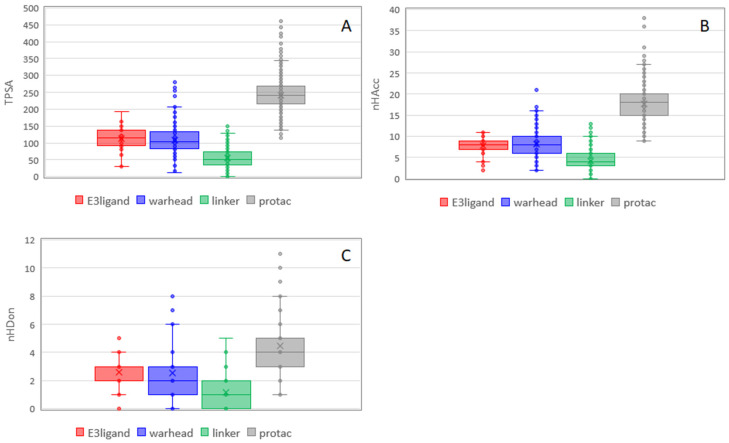
Box plot of the polar descriptors for PROTACs and their building blocks. (**A**) TPSA, (**B**) nHAcc, and (**C**) nHDon. Statistical analysis of data was performed with ANOVA. *p*-values are reported in Appendix A.

**Figure 8 molecules-26-00672-f008:**
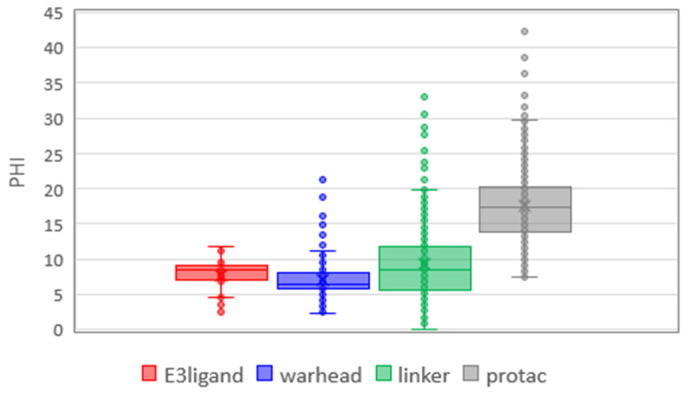
Box plot of the Kier’s flexibility index (PHI). Statistical analysis of data was performed with ANOVA. *p*-values are reported in Appendix A.

**Figure 9 molecules-26-00672-f009:**
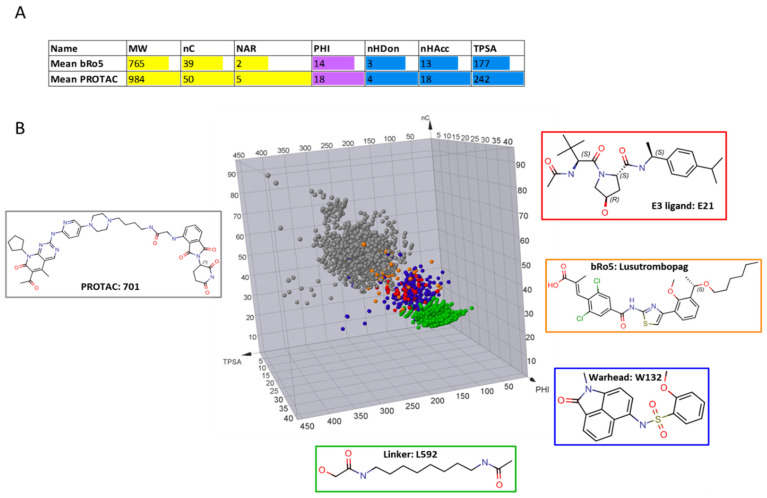
PROTACs vs bRo5, data taken from [13]. (**A**) Mean value of the calculated descriptors; (**B**) 3D plot: nC, TPSA, and PHI were selected as descriptors of the nonpolar, polar, and flexible components, respectively, of the molecular structures. The chosen colors were the following: PROTACs = grey, bRo5 = orange, linkers = green, E3 ligands = red, and warheads = blue. A representative structure for each class is shown for comparison.

**Figure 10 molecules-26-00672-f010:**
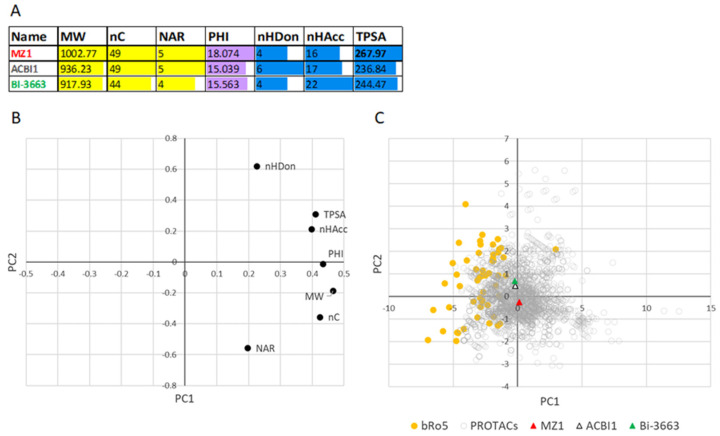
(**A**) Mean descriptor values for three PROTAC with low (MZ1), average (ACBI1), and high (BI-3663) Caco-2 log P_app_ [20] (chemical structures in Appendix A); (**B**) loading plot, and (**C**) score plot. Color code: PROTACs = grey, bRo5 = orange, MZ1 = red, ACBI1 = white and BI-3663 in green.

**Figure 11 molecules-26-00672-f011:**
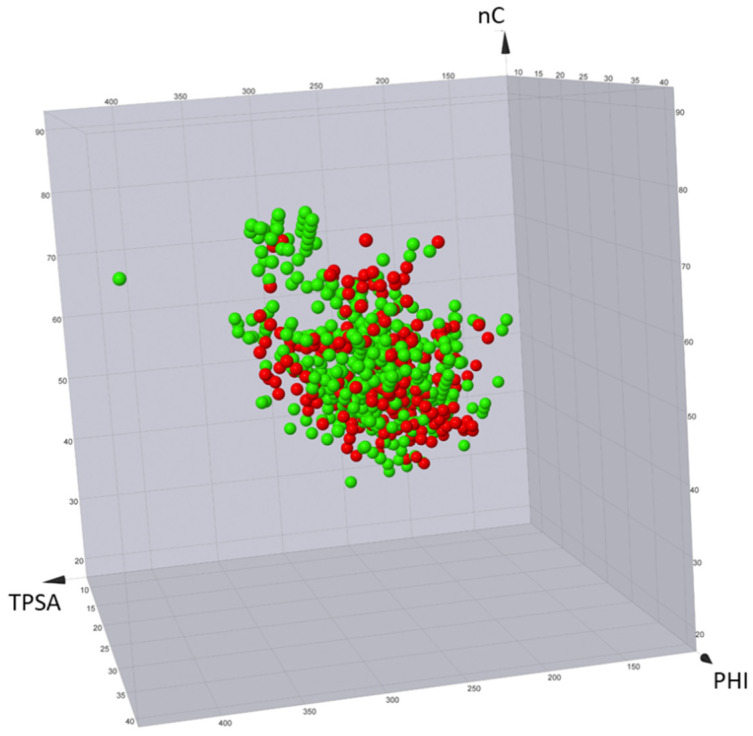
3D plot of active (green) vs. inactive (red) PROTACs obtained from PROTACpedia. nC, TPSA, and PHI were selected as descriptors of the nonpolar, polar, and flexible components, respectively, of the molecular structures.

## Data Availability

The data presented in this study are available on request from the corresponding author.

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
