# Peer review of "PROTACs and Building Blocks: The 2D Chemical Space in Very Early Drug Discovery"

_molecules, 2021, doi:10.3390/molecules26030672_

Round 1

Reviewer 1 Report

The manuscript by Garon et al. is focused on the calculation of 2D descriptors of degrader molecules (as well as linkers, POI ligands and E3 ligase ligands), followed by performing various correlations, PCAs and plots with the aim to characterize the chemical space encompassed by degraders.

In general, I do agree that an update of the last two studies is needed, especially when backed up by two databases that enable the assembly of a more thorough and complete list of degraders. I also agree with authors that sets of descriptors to thoroughly describe heterobifunctional molecules are beneficial in many aspects during design of degraders for a given target (especially for targets without active PROTACs known).

There are two major issues with this manuscript that need to be addressed:

  1. Despite authors’ statements regarding logP values being less reliable is somewhat correct, because you see a discrepancy between calculated and measured values (even in our own lab we experienced this), I believe that in a series of similar degraders, a relative comparison of calculated logP values between degraders utilizing the same POI ligand and the same E3 ligase can be very beneficial. Especially when you want to distinguish between active PROTAC (causing degradation) and inactive one, be it due to differences in permeability of lack of ternary complex formation. Therefore, I am certain that including calc. logP values would be beneficial for this study.
  2. More importantly, the PROTAC-DB includes all degraders published, ranging from inactive compounds to very potent degraders. Therefore, it would make much more sense to divide these two sets of compounds and then make calculations of descriptors. The majority of researchers in this field would like to very much know why one PROTAC works and why one with only slightly different structure is completely inactive – could this be pinpointed by differences in their descriptors? We cannot make this assumption from this study because all degraders were ‘put in the same basket’.

I guess an even better option for calculating parameters properly would be if degraders within a single target would be analyzed (again divided into active and inactive PROTACs) and the results from multiple such analyses compared.

Other minor issues:

In lines 336 and 337 it is stated that in ref 17. a preference for Caco-2 cell permeability data versus PAMPA is preferred – this is not true as in ref 17. this is not described nor mentioned.

Some of the conclusions of the authors seem very logical and would not need to be emphasized in such details (section 2.4).

Text from line 235 to 262 is very difficult to understand.

Author Response

Reviewer 1

The manuscript by Garon et al. is focused on the calculation of 2D descriptors of degrader molecules (as well as linkers, POI ligands and E3 ligase ligands), followed by performing various correlations, PCAs and plots with the aim to characterize the chemical space encompassed by degraders.

In general, I do agree that an update of the last two studies is needed, especially when backed up by two databases that enable the assembly of a more thorough and complete list of degraders. I also agree with the authors that sets of descriptors to thoroughly describe heterobifunctional molecules are beneficial in many aspects during the design of degraders for a given target (especially for targets without active PROTACs known).

We appreciated that this Reviewer valued the aim of our paper and the contribution we provide to help the discovery of new PROTACs. In the revised manuscript all the changes are colored in red.

There are two major issues with this manuscript that need to be addressed:

  1. Despite authors’ statements regarding logP values being less reliable is somewhat correct, because you see a discrepancy between calculated and measured values (even in our own lab we experienced this), I believe that in a series of similar degraders, a relative comparison of calculated logP values between degraders utilizing the same POI ligand and the same E3 ligase can be very beneficial. Especially when you want to distinguish between active PROTAC (causing degradation) and inactive one, be it due to differences in permeability of lack of ternary complex formation. Therefore, I am certain that including calc. logP values would be beneficial for this study.

We understand that removing log P from the pool of descriptors is somewhat unusual. However, this is needed as explained in the text. We agree with the Reviewer that comparing log P in series could be beneficial (and thus we commented this in the revised version) but since we refer to specific situations, also this comment indirectly supports our choice of not including log P in the set of considered descriptors. Finally, our decision was explicitly supported by another reviewer.

  1. More importantly, the PROTAC-DB includes all degraders published, ranging from inactive compounds to very potent degraders. Therefore, it would make much more sense to divide these two sets of compounds and then make calculations of descriptors. The majority of researchers in this field would like to very much know why one PROTAC works and why one with only slightly different structure is completely inactive – could this be pinpointed by differences in their descriptors? We cannot make this assumption from this study because all degraders were ‘put in the same basket’.

I guess an even better option for calculating parameters properly would be if degraders within a single target would be analyzed (again divided into active and inactive PROTACs) and the results from multiple such analyses compared.

These are good points, and we added a section in the end of the Results (subheading 2.6) to properly address them. Shortly, to include activity in the discussion, you need comparable activity values for a good number of PROTACs and this is not feasible using the PROTAC-DB since listed values are obtained through very different assays (biochemical, biophysical and cellular). Since in PROTACpedia the authors report an activity flag (active/inactive), we used it and results are discussed in the text.

Other minor issues:

In lines 336 and 337 it is stated that in ref 17. a preference for Caco-2 cell permeability data versus PAMPA is preferred – this is not true as in ref 17. this is not described nor mentioned.

Line 357. Right. The text has been modified accordingly.

Some of the conclusions of the authors seem very logical and would not need to be emphasized in such details (section 2.4).

The topic investigated in this paper is rather new and many researchers are not familiar with this type of analysis. Therefore, we do believe that it is better to provide a detailed discussion rather than shorten the text and lose some passages.

Text from line 235 to 262 is very difficult to understand.

Lines 254-282. In the mentioned lines we reported a classical way to discuss PCA results. However, to address this reviewer’s comment and make clearer the text, we added some definitions (score and loading plots and variance).

Reviewer 2 Report

GENERAL COMMENTS: The authors present a study of the chemical space, according to 7 different descriptors, of PROTACs and their corresponding building blocks. There is a good introduction to the topic and contextualisation with the related literature, where authors clearly justify the need for this work. They carry out some trend analyses for those 7 descriptors which reveal interesting underlying patterns. Additionally they provide a small validation at the end, where they put the analysis in context with compounds with experimentally determined permeability. This study is interesting and relevant in nature but I find it biased due to two main issues (listed below) which would require some considerable additional work beyond modification of the text in the manuscript. In this review I provide in depth comments with the genuine intention to help the authors produce a good quality manuscript.

Main issue: The analysis is based on PROTAC-DB data exclusively and it is not clear to me why data from a second database (PROTACpedia) was not used as well. From the database’s website I see it is possible to download the full database (upon free registration) so I would say the authors must give a strong reason for not including all available data (perhaps there is complete overlap between the two databases?). If such reason does not exist, I would say it is imperative that the authors re-run all analyses with the data from both datasets.

Second issue: I think that, considering we now have dozens of interpretable descriptors and molecular fingerprints freely available, using just 7 physchem descriptors comes accross as arbitrary. For example why consider the number of carbons and not the number of any other atoms? This type of study should be data-driven as much as possible and selecting a small arbitrary set of descriptors may be limiting the insight one draws from the data. Some possibly interesting descriptors to include could be ring count, aliphatic-to-aromatic atom ratio, single bond count, double bond count, number of polar atoms, ratio of polar-to-non polar atoms. Additionally, other interesting descriptors could be the presence of different substructures of interest e.g. presence and/or count of amines, carboxyl groups, etc. Other classical analyses such as the diversity of murcko scaffolds and an analysis of closest neighbors in non-PROTACS data (e.g. ChEMBL compounds) should be done to ascertain how different is this set of compounds to typical bioactives. I would also suggest an analysis of maximum common substructures. I would be interested in seeing whether different classes share, for example, any Murcko scaffolds and what’s the overlap. These (and other posssible) analyses would be essential to make this study more meaningful.

------

Minor issues:

It is concerning that the methodology section has only 6 lines and misses several details that would be required to properly reproduce this study. How were the structures prepared prior to descriptor calculation? If no structure standardisation was employed, this should be done and descriptors recalculated with new structures. Were the descriptors normalised prior to the PCA analysis? If yes, the type of normalisation should be specified; if no, this is essential prior to carrying out the PCA. The production of each plot should be described in detail such as what function was used in the software. What function was used to produce the different scatter plots (both the 2D and 3D ones) and the box plots?

I recommend going back and reading carefully the section explaining how descriptors were selected. The authors state LogP is not accurate and discard it, however they also state TPSA is not accurate for bRo5 compound but still decide to use it. It is clear that TPSA is problematic because these are typically much larger compounds than a typical bioactive compound and actual solvent accessibility can differ greatly depending on the conformation. However the same can be said for Hydrogen donors and acceptors. So, if the authors decide to discuss the shortcomings of TPSA and LogP this has to be done in the same proportion for all other descriptors when applicable.

Regarding the plots: please replace the name “box a whisker plots” with simply “box plots”. The authors should go through all plots and make sure information is clear. For instance, Figure 6 does not allow to clearly identify the median line and the upper and lower lines of the interquartile extremes, as all elements in the box plots have the same color. Secondly, if the authors emphasize that the 3 classes of compounds have 3 particular colors, these colors should be reserved to represent these compounds. For example, Figure 10 A misleads the reader into thinking MZ1 belongs to the E3 ligands and BI-3663 belongs to the linkers because they are highlighted in red and green, correspondingly. For many of the box plots as much of the plot is occupied with white space as it is with the actual box plots, please trim the "empty" spaces. Legends and numbers are also typically too small - please make these larger.

The study has no statistical tests associated with the different correlations/distributions shown which is essential to establish actual difference beyond apparent visual differences. The authors do not specify what type of correlation function was used in their correlation matrices (which should be clarified), but I am asuming these matrices are populated with the Pearson’s correlation coeficient (R). In this case it might be useful to look at the Spearman’s R as this is appropriate to evaluate the correlation between variables with different scale and/or magnitude.

Specific corrections:

Figure 1 has incorrect placement of “B” and “C”. I would suggest using a image editing tool - there are different free tools to do this very easily such as Gimp.

Line 17: Replace “matrixes” with “matrices” (matrices is the correct plural of matrix).

Line 38: Remove the accronym “POI” and just state protein of interest. It is used one one other time in the whole text.

Line 38: you should mention that these three components are also known as building blocks, given you use this name throughout.

Line 41: Please provide the meaning of the accronym “PROTAC”, which is PROteolysis TArgeting Chimera.

Line 42: The term “suffer” is more appropriate when refering to, for example, a patient. Replace “suffer” with “exhibit” or something equivalent. Remove the “DMPK” accronym as it is used only one other time.

Line 61: Provide the correct name of the database: Instead of “the Weizmann

PROTAC-DB” you should have “PROTACpedia”.

Line 63-64: “shown to give an impulse in the development” is a bit of a strange phrase. Perhaps replace with “... shown to promote the development...”.

Line64-65: Regarding the phrase “... perhaps the most known example of success...”, success here is a generic descriptor. Success in what, exactly? Perhaps you could simply say That PDB, ChemSpider, ZINC, etc, are among the most popular databases used in research.

Line 92: I would not call PubChem as bioinformatics resource - it is a cheminformatics resource, rather.

Line 94: Give the precise name of the database (i.e. PROTECpedia), as it is not called “Weizmann DB”.

Line 98: By “and the literature” do you mean “and the corresponding references”?

Line 107: Replace “for any class” with “for every class”.

Line 112-117: I would put this information in a table, providing target coverage consistently. For example I would show the top 10 targets for each class, with a column with the target name and/or its uniprotID, and a column with the compound count. Just because there are many targets in the warheads class does not mean you should list just 3 of them. Also, the authors sometimes appear to collapse targets into the same entry (e.g. RNF) which is listed alongside single targets. I would rather present “RNF Family (RNF114, RNF4)” or just present them separately, because at first sight, from the list in these line, I would think RNF is a target just like cIAP1. Additionaly, the wuthors should make sure there is consistency in the way targets are defined: for example cIAP1 is not defined but MDM2 is. I would also like to see the total number of targets covered per each class of compounds.

Line 140: I am not sure “drug-related compounds” is the correct term. Perhaps replace with drug-like compounds.

Line 142: replace “very-early” with “very early”.

Line 143: replace “allows their understanding and practical application by medicinal chemists” with “allows straightforward interpretation and use by medicinal chemists”.

Line 144: Like stated above, the authors should avoid stating that molecules “suffer from” - rather, please use “exhibit drawbacks” or “are associated with drawbacks”.

Line 151: either remove ’of interest’ or keep these words outside quotations. Replace “later than” with “in a later stage of development”.

Line 153: I am not sure that the authors mean by “once some real samples are available”. Dos this mean “once compounds are synthesised and experimentally tested”? Please make sentence clearer.

Line 154: Please replace the sentence in parenthesis with “mostly obtained through chromatography due to suboptimal compound purity”.

Line 158-163: It is interesting and very relevant that LogP has not been taken at face value, as it is usually done, and I do agree with the authors’ decision to not include it seeinf as it has shown poor applicability to these larger molecules.

Line 164-165: It is incorrect to asume TPSA is generally interpreted by researchers as “the maximum polarity expressed by a compound”. I am sure most people are aware that TPSA, as the name indicates, measures the (topological) polar surface area of a compound. Therefore the authors should remove the sentence “which is a 2D value of the maximum polarity that can be expressed by a compound”. Additionally, in order to say this is not a good predictor of polarity of the bRo5, here pertaining to PROTACs, the authors must provide some scientific evidence (either a reference or their own analyses) showing that TPSA or a related descriptor is miscalculated for PROTACs and PROTAC-related compounds. Otherwise, I would recommend a more conservative statement “TPSA is not expected to reliably represent these larger compounds”.

Line 167: I believe the correct nomenclature for “fully exposed on the surface or masked” is “solvent-accessible or not”.

Line 169: The authors cannot spend 5 lines basically arguing why TPSA is a inacurate descriptor for bRo5, and end up saying they will use TPSA after all, because it is informative when used with 3D PSA (which they didn’t even use in this work). It comes across as the authors are contradicting themselves. I would suggest presenting a more conservative statement that shows the reader that the authors are aware TPSA is limited particularly for these larger compounds, but they are still using it because it, in the least, provides a rough picture of the ratio between polar and non-polar group across the molecule.

Line 175: Stating that some software packages do not describe how they calculate HDB/HBA has no bearing in this paper as it says nothing about the validity of this property as a descriptor. There are both good and bad packages available for use, and the typically used ones e.g. RDKit, openbabel, MOE, describe exactly in their documentation what they consider for these counts. The reader will asume the highest quality methodology was used by the authors, so there is no need to state that there are some imprecisions/inconsistencies in some packages generally available. This type of discussion belongs in a paper reviewing different packages and their level of agreement. I would rather the authors had discussed the shortcomings of applying typical hydrogen bonding counts to PROTACS. Perhaps a line or two discussing whether some highly flexible compounds may achieve a conformation that renders some hydrogen bonding groups no longer available to establish a bond, or whether flexibility has no expected effect on hydrogen bonding capacity whatsoever. I would expect this to have an effect given the sentence that the authors have earlier in the text saying that “a fragment [may be] fully exposed on the surface or masked”.

Line 209 (but also applicaple throughout): Please provide more informative and specific subheadings throughout. For instance “Building blocks and PROTACs analysis” should be replaced with something like “Analysis of chemical space distribution of PROTACs and their building blocks”. Additionally make sure subheadings are both consistent and non-overlapping. You have “2.3 Building blocks and PROTACs analysis” and then “2.4 Building blocks vs PROTACs”. To me this reads as the same.

Line 345: In order to come across as cherry-picking, the authors should justify why were these 3 compounds selected. Are they the only 3 with permeability values? If yes, this should be mentioned. If there are more compounds, they should either all be included in the analysis of the selection criteria should be clearly stated.

Line 349: “Excessive flexibility is often associated with poor permeability.” Even though this is semi-obvious, I think a reference is needed.

Author Response

Reviewer 2.

GENERAL COMMENTS: The authors present a study of the chemical space, according to 7 different descriptors, of PROTACs and their corresponding building blocks. There is a good introduction to the topic and contextualisation with the related literature, where authors clearly justify the need for this work. They carry out some trend analyses for those 7 descriptors which reveal interesting underlying patterns. Additionally, they provide a small validation at the end, where they put the analysis in context with compounds with experimentally determined permeability. This study is interesting and relevant in nature but I find it biased due to two main issues (listed below) which would require some considerable additional work beyond modification of the text in the manuscript. In this review I provide in depth comments with the genuine intention to help the authors produce a good quality manuscript.

We sincerely appreciated the great work made by this Reviewer to provide suggestions/comments intended to improve the quality of the manuscript. Additional work has been performed and the text has been significantly revised. In the revised manuscript all the changes are colored in red.

Main issue: The analysis is based on PROTAC-DB data exclusively and it is not clear to me why data from a second database (PROTACpedia) was not used as well. From the database’s website I see it is possible to download the full database (upon free registration) so I would say the authors must give a strong reason for not including all available data (perhaps there is complete overlap between the two databases?). If such reason does not exist, I would say it is imperative that the authors re-run all analyses with the data from both datasets.

Good point. In the revised version we provided more room for PROTACpedia (we also contacted the database main author to get additional explanations about the server). In particular, after downloading all data, we verified in Fig. S1 the very large overlap with PROTAC-DB, whereas in the new Section 2.6 we used PROTACpedia to discuss activity data.

Second issue: I think that, considering we now have dozens of interpretable descriptors and molecular fingerprints freely available, using just 7 physchem descriptors comes accross as arbitrary. For example why consider the number of carbons and not the number of any other atoms? This type of study should be data-driven as much as possible and selecting a small arbitrary set of descriptors may be limiting the insight one draws from the data. Some possibly interesting descriptors to include could be ring count, aliphatic-to-aromatic atom ratio, single bond count, double bond count, number of polar atoms, ratio of polar-to-non polar atoms. Additionally, other interesting descriptors could be the presence of different substructures of interest e.g. presence and/or count of amines, carboxyl groups, etc. Other classical analyses such as the diversity of murcko scaffolds and an analysis of closest neighbors in non-PROTACS data (e.g. ChEMBL compounds) should be done to ascertain how different is this set of compounds to typical bioactives. I would also suggest an analysis of maximum common substructures. I would be interested in seeing whether different classes share, for example, any Murcko scaffolds and what’s the overlap. These (and other possible) analyses would be essential to make this study more meaningful.

We understand the Reviewer's point of view, but we have a different opinion. In this paper, we are interested in providing a univocal starting point in the definition of the degraders’ chemical space. To reach this aim we have chosen a small set of simple 2D descriptors that are clearly related to the physicochemical properties that in principle govern fundamental properties involved in DMPK profiling. It is true that their choice is arbitrary, but we provide strong arguments for it. As outlined in the text, it is evident that the number of descriptors is expected to be increased in the near future once more experimental data will be available. For instance, new determinants able to consider the potential of degraders to form intramolecular hydrogen bonds are expected to be included in an updated version of the descriptors. In practice, we prefer to apply a constructionist rather than a reductionist approach often associated with the inclusion of too many not completely clear descriptors. Furthermore, the Murcko analysis, and similar methodologies, were set up to catch analogies and differences in the common scaffolds present in very heterogeneous datasets of drugs, for example, Murcko analyzed a set of about 5000 drugs that in 1996 were included in the Comprehensive Medicinal Chemistry (CMC) book. In our opinion, at this stage of available knowledge of PROTACs, the number of scaffolds used in the three blocks is not so high. This suggests exploring how the physicochemical properties of the blocks influence the final entity instead of performing a scaffold analysis. In conclusion, we think that the study of the popularity of certain scaffolds with respect to others will be interesting when the number of POI ligands and E3 ligases will be greater. But a very meaningful breakthrough is expected when the experimental properties, such as lipophilicity and permeability, will be available for a large number of PROTACs. These data will allow to aggregate scaffolds on the basis of properties that govern their future as drugs. Finally, we would like to point out that we already successfully applied a similar approach in other research fields (i.e. identification of essential physicochemical properties governing chromatographic indexes) by grouping a set of descriptors into blocks (e.g. Caron et al. Curr. Pharm. Des., 2020; 26(44):5662-5667). In conclusion, we think that our approach is able to extract the main information about PROTACs on the basis of available information. We are sure that the increasing interest in this class of compounds will soon lead to the release of a big amount of experimental data that will allow understanding the main factors that influence non only the protein degradation activity but also DMPK properties of PROTACs.

------

Minor issues:

It is concerning that the methodology section has only 6 lines and misses several details that would be required to properly reproduce this study. How were the structures prepared prior to descriptor calculation? If no structure standardisation was employed, this should be done and descriptors recalculated with new structures. Were the descriptors normalised prior to the PCA analysis? If yes, the type of normalisation should be specified; if no, this is essential prior to carrying out the PCA. The production of each plot should be described in detail such as what function was used in the software. What function was used to produce the different scatter plots (both the 2D and 3D ones) and the box plots?

We improved the Methods section. However, since only 2D descriptors were calculated, structures downloaded as .sdf files do not require any specific preparation. To perform PCA we used commercial software that implements data standardization which makes the values of each feature in the data have zero-mean and unit-variance. In our previous papers including PCA we were never asked to specify the standardization process details.

I recommend going back and reading carefully the section explaining how descriptors were selected. The authors state LogP is not accurate and discard it, however they also state TPSA is not accurate for bRo5 compound but still decide to use it. It is clear that TPSA is problematic because these are typically much larger compounds than a typical bioactive compound and actual solvent accessibility can differ greatly depending on the conformation. However the same can be said for Hydrogen donors and acceptors. So, if the authors decide to discuss the shortcomings of TPSA and LogP this has to be done in the same proportion for all other descriptors when applicable

We modified section 2.1 introducing a clearer analysis of our choice to focus on a small set of 2D descriptors.

Regarding the plots: please replace the name “box a whisker plots” with simply “box plots”. The authors should go through all plots and make sure information is clear. For instance, Figure 6 does not allow to clearly identify the median line and the upper and lower lines of the interquartile extremes, as all elements in the box plots have the same color. Secondly, if the authors emphasize that the 3 classes of compounds have 3 particular colors, these colors should be reserved to represent these compounds. For example, Figure 10 A misleads the reader into thinking MZ1 belongs to the E3 ligands and BI-3663 belongs to the linkers because they are highlighted in red and green, correspondingly. For many of the box plots as much of the plot is occupied with white space as it is with the actual box plots, please trim the "empty" spaces. Legends and numbers are also typically too small - please make these larger.

The reviewer is right. All plots have been modified accordingly.

The study has no statistical tests associated with the different correlations/distributions shown which is essential to establish actual difference beyond apparent visual differences. The authors do not specify what type of correlation function was used in their correlation matrices (which should be clarified), but I am asuming these matrices are populated with the Pearson’s correlation coeficient (R). In this case it might be useful to look at the Spearman’s R as this is appropriate to evaluate the correlation between variables with different scale and/or magnitude.

In the paper, we provided graphical analyses associated with a visual inspection. In our opinion, in this case, statistical data do not improve the amount of provided information. We used Pearson’s correlation coefficient (now specified in the text, line 418) since we are looking for linear correlations.

Specific corrections:

Figure 1 has incorrect placement of “B” and “C”. I would suggest using a image editing tool - there are different free tools to do this very easily such as Gimp.

We made some mistakes importing the figure in the World file, now we have amended the figure (figure 3 in the revised manuscript).

Line 17: Replace “matrixes” with “matrices” (matrices is the correct plural of matrix).

Done in all the manuscript

Line 38: Remove the accronym “POI” and just state protein of interest. It is used one one other time in the whole text.

Line 39 in the revised form. Done.

Line 38: you should mention that these three components are also known as building blocks, given you use this name throughout.

Line 39. Done

Line 41: Please provide the meaning of the accronym “PROTAC”, which is PROteolysis TArgeting Chimera.

Done

Line 42: The term “suffer” is more appropriate when refering to, for example, a patient. Replace “suffer” with “exhibit” or something equivalent. Remove the “DMPK” accronym as it is used only one other time.

Line 49. We replaced “suffer” in the text. It is true that the acronym DMPK is used only one time but it is a standard, widely accepted definition and in our opinion, it can help the reader to better understand the paper research field.

Line 61: Provide the correct name of the database: Instead of “the Weizmann PROTAC-DB” you should have “PROTACpedia”.

Line 67. We amended the text.

Line 63-64: “shown to give an impulse in the development” is a bit of a strange phrase. Perhaps replace with “... shown to promote the development...”.

Line 70. Done

Line64-65: Regarding the phrase “... perhaps the most known example of success...”, success here is a generic descriptor. Success in what, exactly? Perhaps you could simply say That PDB, ChemSpider, ZINC, etc, are among the most popular databases used in research.

Lines 73-74. We modified the text to better explain the concept

Line 92: I would not call PubChem as bioinformatics resource - it is a cheminformatics resource, rather.

Line 178. It is right, done

Line 94: Give the precise name of the database (i.e. PROTECpedia), as it is not called “Weizmann DB”.

Line 180. Done

Line 98: By “and the literature” do you mean “and the corresponding references”?

Line 184. Ok, done

Line 107: Replace “for any class” with “for every class”.

Line 197. Done

Line 112-117: I would put this information in a table, providing target coverage consistently. For example I would show the top 10 targets for each class, with a column with the target name and/or its uniprotID, and a column with the compound count. Just because there are many targets in the warheads class does not mean you should list just 3 of them. Also, the authors sometimes appear to collapse targets into the same entry (e.g. RNF) which is listed alongside single targets. I would rather present “RNF Family (RNF114, RNF4)” or just present them separately, because at first sight, from the list in these line, I would think RNF is a target just like cIAP1. Additionaly, the wuthors should make sure there is consistency in the way targets are defined: for example cIAP1 is not defined but MDM2 is. I would also like to see the total number of targets covered per each class of compounds.

We agree with this comment. Table S2 and S3 now report a deep analysis of the database entries. This required a lot of work, but it is true that previously we provided a rather superficial overview of the database.

Line 140: I am not sure “drug-related compounds” is the correct term. Perhaps replace with drug-like compounds.

Line 98. It is right, drug-like is better.

Line 142: replace “very-early” with “very early”.

Line 100. Ok

Line 143: replace “allows their understanding and practical application by medicinal chemists” with “allows straightforward interpretation and use by medicinal chemists”.

Line 101. Ok

Line 144: Like stated above, the authors should avoid stating that molecules “suffer from” - rather, please use “exhibit drawbacks” or “are associated with drawbacks”.

Line 102. Ok

Line 151: either remove ’of interest’ or keep these words outside quotations. Replace “later than” with “in a later stage of development”.

Line 109. Ok

Line 153: I am not sure that the authors mean by “once some real samples are available”. Dos this mean “once compounds are synthesised and experimentally tested”? Please make sentence clearer.

Lines 111-112. We have modified the sentence.

Line 154: Please replace the sentence in parenthesis with “mostly obtained through chromatography due to suboptimal compound purity”.

Line 113. Ok

Line 158-163: It is interesting and very relevant that LogP has not been taken at face value, as it is usually done, and I do agree with the authors’ decision to not include it seeinf as it has shown poor applicability to these larger molecules.

We appreciate this comment because the problem is very often underestimated. We have discussed this point in other papers where we showed how Log P calculators are not reliable when applied to large molecules

Line 164-165: It is incorrect to asume TPSA is generally interpreted by researchers as “the maximum polarity expressed by a compound”. I am sure most people are aware that TPSA, as the name indicates, measures the (topological) polar surface area of a compound. Therefore the authors should remove the sentence “which is a 2D value of the maximum polarity that can be expressed by a compound”. Additionally, in order to say this is not a good predictor of polarity of the bRo5, here pertaining to PROTACs, the authors must provide some scientific evidence (either a reference or their own analyses) showing that TPSA or a related descriptor is miscalculated for PROTACs and PROTAC-related compounds. Otherwise, I would recommend a more conservative statement “TPSA is not expected to reliably represent these larger compounds”.

Line 167: I believe the correct nomenclature for “fully exposed on the surface or masked” is “solvent-accessible or not”.

Line 169: The authors cannot spend 5 lines basically arguing why TPSA is a inacurate descriptor for bRo5, and end up saying they will use TPSA after all, because it is informative when used with 3D PSA (which they didn’t even use in this work). It comes across as the authors are contradicting themselves. I would suggest presenting a more conservative statement that shows the reader that the authors are aware TPSA is limited particularly for these larger compounds, but they are still using it because it, in the least, provides a rough picture of the ratio between polar and non-polar group across the molecule.

The section about TPSA has been modified to take into accounts all these comments. Overall, we shortened the discussion about TPSA (starting from line 134) to avoid appearing to contradict ourselves (a good point raised by the Reviewer). We motivated its selection as a polarity descriptor since TPSA is a descriptor of the maximum polarity that can be expressed by a compound and as such has a value. We also added two references that confirm this definition, the first by us (10.1016/j.drudis.2016.11.017) and the second by Whitty et al (“For a reasonably flexible molecule, the major conformer in aqueous solution will have all or most polar groups exposed to solvent, such that MPSAAq can be approximated simply by using TPSA”, doi: 10.1016/j.drudis.2016.02.005).

Line 175: Stating that some software packages do not describe how they calculate HDB/HBA has no bearing in this paper as it says nothing about the validity of this property as a descriptor. There are both good and bad packages available for use, and the typically used ones e.g. RDKit, openbabel, MOE, describe exactly in their documentation what they consider for these counts. The reader will asume the highest quality methodology was used by the authors, so there is no need to state that there are some imprecisions/inconsistencies in some packages generally available. This type of discussion belongs in a paper reviewing different packages and their level of agreement. I would rather the authors had discussed the shortcomings of applying typical hydrogen bonding counts to PROTACS. Perhaps a line or two discussing whether some highly flexible compounds may achieve a conformation that renders some hydrogen bonding groups no longer available to establish a bond, or whether flexibility has no expected effect on hydrogen bonding capacity whatsoever. I would expect this to have an effect given the sentence that the authors have earlier in the text saying that “a fragment [may be] fully exposed on the surface or masked”.

In this case, we do not agree with the reviewer. First, explicit definition of HBA and HBD is rarely reported in the literature and, honestly, we never read warnings about it. Therefore, we prefer to keep the sentence about software that does not distract the reader from the focus of the paper but suggests some precaution to adopt. Second, once defined in the beginning that we are focusing on 2D descriptors and discussed their limits, it is useless repeating the concept for any descriptor.

Line 209 (but also applicaple throughout): Please provide more informative and specific subheadings throughout. For instance “Building blocks and PROTACs analysis” should be replaced with something like “Analysis of chemical space distribution of PROTACs and their building blocks”. Additionally make sure subheadings are both consistent and non-overlapping. You have “2.3 Building blocks and PROTACs analysis” and then “2.4 Building blocks vs PROTACs”. To me this reads as the same.

Excellent comment. We modified the text accordingly.

Line 345: In order to come across as cherry-picking, the authors should justify why were these 3 compounds selected. Are they the only 3 with permeability values? If yes, this should be mentioned. If there are more compounds, they should either all be included in the analysis of the selection criteria should be clearly stated.

Lines 369-370. OK, the text has been modified accordingly.

Line 349: “Excessive flexibility is often associated with poor permeability.” Even though this is semi-obvious, I think a reference is needed.

Line 372. Done.  

Reviewer 3 Report

General: This paper takes a data science approach to analyze the physiochemical properties of proteolysis targeting chimeras (PROTACs). Unlike traditional enzymatic inhibitors which are usually comprised of a monovalent warhead, PROTACs tend to be heterobifunctional, consisting of a flexible linker, a E3 ligase targeting moiety, and a warhead. Therefore, it would not be unexpected that they would have profoundly different physiochemical profiles that could influence their pharmacodynamics and bioavailability. Unlike most orally bioavailable drugs PROTACs do not follow Lipinski’s rule of 5. However, there are some drugs that do not follow those rules and still show good bioavailability. The authors show some drugs that do not follow Lipinski’s rule of 5 overlap some of the same chemical space as some PROTACs.

Specific:

  • For Figures 6a-c, Figures 7a-c, and Figure 8 the authors should show a comparison with other pharmacological inhibitors and bRo5 drugs.
  • For Figure 9, authors should show a representative structure from each class of drugs for comparison.
  • The overview of the databases and datasets, while extremely interesting, would be more appropriate in a review of publicly accessible information about PROTACs.

Author Response

Reviewer 3.

General: This paper takes a data science approach to analyze the physiochemical properties of proteolysis targeting chimeras (PROTACs). Unlike traditional enzymatic inhibitors which are usually comprised of a monovalent warhead, PROTACs tend to be heterobifunctional, consisting of a flexible linker, a E3 ligase targeting moiety, and a warhead. Therefore, it would not be unexpected that they would have profoundly different physiochemical profiles that could influence their pharmacodynamics and bioavailability. Unlike most orally bioavailable drugs PROTACs do not follow Lipinski’s rule of 5. However, there are some drugs that do not follow those rules and still show good bioavailability. The authors show some drugs that do not follow Lipinski’s rule of 5 overlap some of the same chemical space as some PROTACs.

 In the revised manuscript all the changes are colored in red.

Specific:

  • For Figures 6a-c, Figures 7a-c, and Figure 8 the authors should show a comparison with other pharmacological inhibitors and bRo5 drugs.

These plots were designed to highlight PROTACs and building blocks properties and thus they do not need additional data. Moreover, a comparison with bRo5 compounds is provided in Fig. 9.

  • For Figure 9, authors should show a representative structure from each class of drugs for comparison.

Done.

  • The overview of the databases and datasets, while extremely interesting, would be more appropriate in a review of publicly accessible information about PROTACs.

We do not understand this comment. We described the databases that we used to extract the compounds on which our analysis was performed.

Round 2

Reviewer 1 Report

The authors made significant changes to the paper and touched upon the issues raised carefully. Just need revise minor spell-checking mistakes that will be solved during proofreading. In this case, I recommend the paper to be published.

Author Response

No reply is needed 

Reviewer 2 Report

The authors addressed most of the corrections and comments, which I appreciate. One main issue remains unaddressed regarding the matter of the amount and type of descriptors used.

I agree with the authors that all selected descriptors are appropriately justified. I also agree that there are many descriptors of questionable applicability to molecules such as PROTACs. However this still does not justify selecting, say, the number of carbons, and leaving out other count-based descriptors which are applicable to these compounds. Why should the authors analyse the carbon atom count and not analyse the polar atom count, or the number of oxygens, or the ratio of polar-to-nonpolar atom count, etc? The author’s reasoning for not including additional descriptors is just a blanket statement that they “prefer to apply a constructionist rather than a reductionist approach often associated with the inclusion of too many not completely clear descriptors”. I totally agree with this stance but the fact that many descriptors with questionable/unknown applicability to PROTACs may exist does not justify the exclusion of some of the descriptors with the same nature as the ones used (for example the descriptors I listed above). Ultimately my opinion is that if there are additional descriptors whose exclusion from the analysis cannot be justified scientifically, excluding them becomes therefore unnaceptable. For this reason, the decision to use only these particular 7 descriptors renders this study limited and biased. I would recommend the authors go through a list of all publicly available descriptors (starting from the ones I mentioned here) and only exclude the descriptors for which they have a scientific justification for exclusion. If you have a look at the RDKit 2D descriptors (https://www.rdkit.org/docs/GettingStartedInPython.html#list-of-available-descriptors) or Mordred descriptors (https://mordred-descriptor.github.io/documentation/master/descriptors.html) the majority might not be applicable to your study but I would estimate that there are at least another 8-10 descriptors whose exclusion cannot, in good conscience, be justified.

A second issue that remains unaddressed regard the use of statistical tests to support observed trends:

I agree that one is able to demonstrate difference in a plot, but statistics exists because sometimes visual differences are misleading. Without a statistical test one should not say sentences such as “The larger number of aromatic rings of warheads supports the fact that the nonpolar properties of PROTACs mostly depend on them.”. Scientifically speaking, one can one state that X is larger than Y if a statistical test supports this difference. Would the authors still keep this statement if there is no statistical significance?

Author Response

"The authors addressed most of the corrections and comments, which I appreciate. One main issue remains unaddressed regarding the matter of the amount and type of descriptors used.

I agree with the authors that all selected descriptors are appropriately justified. I also agree that there are many descriptors of questionable applicability to molecules such as PROTACs. However this still does not justify selecting, say, the number of carbons, and leaving out other count-based descriptors which are applicable to these compounds. Why should the authors analyse the carbon atom count and not analyse the polar atom count, or the number of oxygens, or the ratio of polar-to-nonpolar atom count, etc? The author’s reasoning for not including additional descriptors is just a blanket statement that they “prefer to apply a constructionist rather than a reductionist approach often associated with the inclusion of too many not completely clear descriptors”. I totally agree with this stance but the fact that many descriptors with questionable/unknown applicability to PROTACs may exist does not justify the exclusion of some of the descriptors with the same nature as the ones used (for example the descriptors I listed above). Ultimately my opinion is that if there are additional descriptors whose exclusion from the analysis cannot be justified scientifically, excluding them becomes therefore unnaceptable. For this reason, the decision to use only these particular 7 descriptors renders this study limited and biased. I would recommend the authors go through a list of all publicly available descriptors (starting from the ones I mentioned here) and only exclude the descriptors for which they have a scientific justification for exclusion. If you have a look at the RDKit 2D descriptors (https://www.rdkit.org/docs/GettingStartedInPython.html#list-of-available-descriptors) or Mordred descriptors (https://mordred-descriptor.github.io/documentation/master/descriptors.html) the majority might not be applicable to your study but I would estimate that there are at least another 8-10 descriptors whose exclusion cannot, in good conscience, be justified."

We understand the comment and the Reviewer’s point of view, but we disagree with her/his opinion about the lack of a scientific procedure in the selection of our pool of 7 descriptors. We decided to adopt a physicochemical approach. The set of descriptors was chosen since related to the two main physicochemical properties (i.e. hydrophobicity and polarity) and a third property related to the flexibility, introduced because it is an important determinant in bRo5 chemical space. We think that there is clear scientific evidence that the molecular weight, the number of carbon atoms, and the number of aromatic rings are related to hydrophobicity whereas TPSA, the number of acceptor and donor hydrogen bonds are related to polarity (and the Kier flexibility index is related to flexibility). We are aware that other descriptors can be related to these properties and in the paper we do not affirm that the selected descriptors are the best or the unique set of descriptors that can be used to analyze the datasets. In particular, we state that when a sufficient number of experimental physicochemical measures will be available, we will use a larger set of descriptors. However, in this paper, we show a descriptive analysis of the datasets on the basis of a supervised scheme based on polarity, hydrophobicity, and flexibility, and in this scheme, we think that the seven selected descriptors are a good choice and other descriptors the inclusion of some of the descriptors with the same nature as the ones used do not add additional information.

Said this, we decided to take advantage of the referee’s opinion and perform an additional effort to satisfy the reviewer. We calculate an additional set of count-based descriptors and we carry out a deeper analysis of the descriptor selection based on statistical methods. Results showed that the physicochemical profile of the PROTACs is caught by the seven descriptors we selected. All the material is now in the SI.

"A second issue that remains unaddressed regard the use of statistical tests to support observed trends: I agree that one is able to demonstrate difference in a plot, but statistics exists because sometimes visual differences are misleading. Without a statistical test one should not say sentences such as “The larger number of aromatic rings of warheads supports the fact that the nonpolar properties of PROTACs mostly depend on them.”. Scientifically speaking, one can one state that X is larger than Y if a statistical test supports this difference. Would the authors still keep this statement if there is no statistical significance?"

We are aware that statistics was developed to avoid artifacts suggested by a simple view, but we do not want to arrive at definitive conclusions because up to now there is no sufficient knowledge about PROTACs to arrive at definitive conclusions. The results of our exploration suggest the presence of some trends.

Said this, we agree that reporting more comprehensive statistical data can clarify some possible doubts in the reader, and thus we reported p-values in SI.

Reviewer 3 Report

The authors adequately revised the manuscript.

No further revisions are necessary. 

Author Response

No reply is needed